# Recognising and responding to the community needs of gay and bisexual men around mpox

John Gilmore[1]*, David Comer[2], David J. Field[3,4‡], Randal Parlour[5‡], Adam Shanley[6‡], Chris Noone[2]

1 School of Nursing, Midwifery and Health Systems, University College Dublin, Dublin, Ireland, 2 School of Psychology, University of Galway, Galway, Ireland, 3 Institute of Applied Health Sciences, University of Aberdeen, Aberdeen, Scotland, 4 Gay Men's Health Service, Health Service Executive, Dublin, Ireland, 5 HSE Health Protection Surveillance Centre, Dublin, Ireland, 6 HIV Ireland, Dublin, Ireland

☯ These authors contributed equally to this work.
‡ DJF, RP and AS also contributed equally to this work.
* john.gimore@ucd.ie

**Data Availability Statement:** Data cannot be shared publicly because of potentially identifiyble data. Data are available from the UCD Ethics Committee research.ethics@ucd.ie for researchers

## Abstract

### Background

In May 2022, a global surge in mpox cases, typically endemic to Western and Central Africa, particularly affected gay, bisexual, and other men who have sex with men (gbMSM). This study examines gbMSM communities' experiences and perceptions around Ireland's public health response to the outbreak.

### Methods

A cross-sectional mixed-methods online survey was conducted. Qualitative data were analysed using reflexive thematic analysis informed by critical realism.

### Findings

A total of 163 gay and bisexual men took part in the survey. Participants accessed information from diverse sources, reporting varying levels of trustworthiness. Overall, participants were well-informed. Four themes were developed from the qualitative data: (1) Perceptions of the mpox response: divergence in urgency, priority, and care; (2) The mpox outbreak as a sign of otherness for gbMSM; (3) The potential for othering through mpox prevention practices; and (4) mpox, memory and fear.

### Discussion

While community-led initiatives were effective, significant challenges included stigmatisation, discrimination, and mistrust towards public health institutions, influenced by institutionalised homophobia. The study underscores the need for inclusive, culturally sensitive, and transparent public health strategies.

who meet the criteria for access to confidential data.

**Funding:** The research was funded by the HSE Sexual Health and Crisis Pregnancy Programme in Ireland. The funders had no role in study design, data collection and analysis, decision to publish, or preparation of the manuscript.

**Competing interests:** The authors have declared that no competing interests exist

## Conclusion

The mpox outbreak highlights the importance of robust community collaboration in public health interventions. Future strategies must ensure equitable access to information, vaccination, and care, and address broader structural inequalities to foster trust and engagement within affected communities.

## Background

In May 2022, a significant surge in globally reported cases of mpox, a viral infection endemic to Western and Central Africa, caught the attention of health authorities worldwide. This marked the first occasion where numerous cases and clusters of mpox were concurrently reported in both endemic and non-endemic regions, spanning Europe and North America [1]. The outbreak was almost exclusively prevalent among communities of gay, bisexual, and other men who have sex with men (gbMSM). By July 2022, the World Health Organization (WHO) had declared the outbreak a Public Health Emergency of International Concern (PHEIC) [2].

The 2022 mpox outbreak raised particular concern among communities of gbMSM) due to the virus's transmission dynamics and the early clustering of cases within this population. mpox transmission occurs through close physical contact with another person with active infection including skin-to-skin contact, which made sexual networks, particularly those with multiple partners, vulnerable to transmission. While mpox is not classified as a sexually transmitted infection, it disproportionately affected gbMSM populations due to the nature of initial outbreak patterns, with intimate gatherings and events contributing to higher transmission rates within the community [3]. In this way it is logical as to why the PHEIC attracted relatively little focus in the general public.

The initial symptoms mimic those of the flu, including fever, low energy, swollen lymph nodes, and general body aches [4]. Usually within one to three days, of fever onset, most infected individuals develop a painful rash or sores that spread across the body. The very visible symptomology at the pustule stage makes the infection quite distinctive and recognisable for those impacted, however, not everyone will display symptoms in this way.

The first mpox case in Ireland was reported on 27 May 2022. By the end of 2022, when data collection for this study concluded, there were 227 confirmed cases, with a noticeable decline in weekly case numbers towards the year's end [5]. To curb the spread of the virus, individuals diagnosed with mpox were advised to self-isolate for up to 30 days.

During the PHEIC, Ireland had specific public health strategies in place, including community outreach programs targeting gbMSM populations, which shaped both the containment of the outbreak and the understanding of its dynamics. In response to the emerging threat, the Health Service Executive (HSE), Ireland's statutory publicly funded healthcare system, established a national crisis management team comprising government officials, public health experts, and representatives from LGBT+ community organisations [6]. This was soon followed by the formation of a strategic advisory group tasked with overseeing the government's public health response.

Given the very specific and targeted impact of the virus on communities of gbMSM, partnership between public health organisations and organisations working within these communities was a key public health strategy [7]. This approach was key in getting information into

communities about prevention strategies and later vaccinations; in many places, this built on connections already established in HIV prevention work [3].

Throughout the public health emergency in Ireland, community organisations played a pivotal role in communication and outreach. The agreed strategy of public health organisations was to facilitate community organisations to lead the information and support efforts. The MPOWER Programme at HIV Ireland and the Man2Man programme administered by the Gay Health Network were instrumental in leading community-focused communication efforts throughout the PHEIC [6].

Despite the absence of a validated mpox vaccine, the smallpox vaccine Imvanex/Jynneos was approved for emergency use under exceptional circumstances globally. However, the roll-out of vaccination programmes faced significant challenges due to a worldwide vaccine shortage. In Ireland, the initial phase of the vaccine rollout targeted individuals deemed to be at highest risk of infection. The eligibility criteria decided upon for this initial phase was limited to those newly diagnosed with early infectious syphilis within the previous six months. By late summer 2022, vaccine availability had improved, allowing broader access for all at-risk individuals. This increase in vaccine supply was partly due to the emergency approval for the intra-dermal administration of Imvanex/Jynneos against mpox, but also increasing global supplies [8].

As part of the response to mpox in Ireland, the MPOWER programme at HIV Ireland was funded to commission research into the gbMSM community needs. This research took the format of a community-based mixed-methods survey.

## Methods

### Project oversight

A project steering group was recruited through the MPOWER programme which included community members, clinicians and public health officials. The group met regularly to oversee the development of the project.

### Participants

We applied convenience sampling through general advertisements in various nightlife venues such as bars, clubs and saunas, as well as on social media; and purposive sampling through LGBTQ+ organisations to recruit participants over 18, identifying as men attracted to men, and living in Ireland. The study included 163 participants, for whom demographic data can be seen in Table 1. The majority of participants identified as gay, were Irish born and living in Dublin. Ireland does not collect national census data on sexual orientation so it is difficult to anticipate how representative the sample is, however a diversity in demographic characteristics were reported by participants.

### Data collection

Following piloting with steering group members, participants self-selected through the Qualtrics survey link on social media between December 6th, 2022, and January 18th, 2023. The survey was shared by HIV Ireland and the MPOWER programme, through paid advertising on Grindr, and through Gay Community News, an Irish LGBTQ+ publication. The survey included a plain language statement and consent procedures, questions eliciting demographic data, and open-ended questions about the impact of mpox on daily life, worries and concerns about prevention, thoughts on the management of mpox, and preferences for information provision. The final section contained closed questions on perceptions of mpox and sources of

**Table 1. Demographic data of participants.**

| Variable | | N | % |
|---|---|---|---|
| Age | 18–24 | 9 | 5.52 |
| | 25–30 | 16 | 9.82 |
| | 31–35 | 40 | 24.54 |
| | 36–40 | 32 | 19.63 |
| | 41–45 | 24 | 14.72 |
| | 46–50 | 19 | 11.66 |
| | 51–55 | 9 | 5.52 |
| | 56–60 | 9 | 5.52 |
| | 61–65 | < 5 | < 3 |
| | 66–70 | < 5 | < 3 |
| Assigned Male at Birth | Yes | 161 | 98.77 |
| | No | < 5 | < 3 |
| Sexual Identity | Gay or homosexual | 139 | 85.28 |
| | Bisexual | 23 | 14.11 |
| | In another way | < 5 | < 3 |
| Ethnicity | Irish | 138 | 84.66 |
| | Irish traveller | < 5 | < 3 |
| | Any other white background | 15 | 9.20 |
| | Chinese | < 5 | < 3 |
| | Any other Asian background | < 5 | < 3 |
| | Latin American | < 5 | < 3 |
| | Any other black background | < 5 | < 3 |
| | Other (including mixed background) | < 5 | < 3 |
| Place of Birth | Argentina | < 5 | < 3 |
| | Armenia | < 5 | < 3 |
| | Brazil | < 5 | < 3 |
| | China | < 5 | < 3 |
| | India | < 5 | < 3 |
| | Ireland | 136 | 83.44 |
| | Italy | < 5 | < 3 |
| | Japan | < 5 | < 3 |
| | Latvia | < 5 | < 3 |
| | Poland | < 5 | < 3 |
| | Portugal | < 5 | < 3 |
| | Russia | < 5 | < 3 |
| | South Africa | < 5 | < 3 |
| | Turkey | < 5 | < 3 |
| | United Kingdom | < 5 | < 3 |
| | United States | < 5 | < 3 |
| Education level | No educational qualifications | < 5 | < 3 |
| | Intermediate/Junior/Group Certificate or equivalent | < 5 | < 3 |
| | Leaving Certificate or equivalent (including Applied Leaving Certificate) | 14 | 8.59 |
| | Higher education below degree level | 25 | 15.34 |
| | Degree or higher | 120 | 73.62 |
| Annual Income | €0–9,999 | < 5 | < 3 |
| | €10,000–19,999 | 15 | 9.20 |
| | €20,000–39,999 | 32 | 19.63 |

(*Continued*)

**Table 1.** (Continued)

| Variable | | N | % |
|---|---|---|---|
| | €40,000–59,999 | 48 | 29.45 |
| | €60,000–99,999 | 50 | 30.67 |
| | €100,000+ | 17 | 10.43 |
| County | Carlow | < 5 | < 3 |
| | Cavan | < 5 | < 3 |
| | Clare | < 5 | < 3 |
| | Cork | 5 | 3.07 |
| | Donegal | < 5 | < 3 |
| | Dublin | 93 | 57.06 |
| | Galway | 10 | 6.13 |
| | Kerry | < 5 | < 3 |
| | Kildare | 9 | 5.52 |
| | Kilkenny | < 5 | < 3 |
| | Laois | < 5 | < 3 |
| | Leitrim | < 5 | < 3 |
| | Limerick | < 5 | < 3 |
| | Mayo | < 5 | < 3 |
| | Meath | 8 | 4.91 |
| | Roscommon | < 5 | < 3 |
| | Tipperary | 5 | 3.07 |
| | Waterford | < 5 | < 3 |
| | Westmeath | < 5 | < 3 |
| | Wexford | < 5 | < 3 |
| | Wicklow | < 5 | < 3 |
| Had mpox | No | 158 | 96.93 |
| | Yes | 5 | 3.07 |

information about mpox. Ethical approval granted by University College Dublin Human Research Ethics Committee. Ref: LS-22-54-Gilmore. Written informed consent was attained through the survey tool by participants confirming their consent to participate in the study and have their data used for research purposes including publication. Data collection was anonymous. Broad demographic data including sexual orientation, geographical region, gender identity, ethnicity, income level and level of education was collected.

## Data analysis

Responses to closed questions were quantitively analysed using Qualtrics, then summarised and visualised using Excel [DC]. We took an ontological position of Critical Realism, interpreting participant responses as a mediated reflection of reality shaped by and embedded within their cultural context [9]. A critical realist ontology recognises the inherent complexity of a social phenomena (as in an outbreak of mpox) in acknowledging the existence of underlying structures and mechanisms that shape the observable events [10]. It provides a framework for understanding how these deep layers of reality interact with human knowledge and experience. Our analysis drew on a socio-ecological model inspired by Baral et al.'s [11] work describing risk contexts for HIV epidemics and the understanding that outbreak stage contextualises risk. We used the six stages of Reflexive Thematic Analysis (RTA) described by Braun & Clarke [9, 12] to analyse qualitative data. Familiarisation required reading and notetaking,

before DC led the initial coding process with JG and CN. Following coding, themes were generated collaboratively through visual mapping of codes and discussion of latent meanings. We developed our own interpretation of themes before collaboratively synthesising and naming them.

## Reflexivity

The principal research team (DC, JG, CN) reflected on our positions as researchers, as queer men, the position of the MPOWER programme as the research sponsor, and the position of the research itself.

Discussions considered the context of the study for gbMSM, on the organisations involved in the mpox response, and on the relationships that we and the MPOWER programme have with the community and relevant organisations. We considered tensions between the responses of participants and the wider public health strategy deployed in this case.

Direct quotations which may be read as derogatory to individuals or organisations have been included to give voice to genuine frustrations expressed by study participants. The inclusion of these quotations are not meant as a reflection of the authors' views, but as a representation of community perceptions expressed in the study.

## Findings

The initial findings presented represent the responses received to quantitative survey questions.

Participants utilised a wide range of sources for information about mpox (see Table 2), with Irish community organisations being the most popular. Almost two-thirds (66.26%) of participants relied on community-led organisations to stay informed about the outbreak. Other frequently used sources included hook-up/dating apps (45.4%), Irish government sources (43.56%), and non-governmental posts on social media platforms like Instagram (31.29%) and Twitter (29.45%). All participants reported accessing information from at least one source, with the number of sources ranging from 1 to 10, and a median of three sources per

**Table 2. Use of sources of information about mpox.**

| Source | | N | % |
|---|---|---|---|
| Media | Irish TV/Radio | 41 | 25.15 |
| | International TV/Radio | 17 | 10.43 |
| Government | Irish government sources (HSE, Department of Health etc.) | 71 | 43.56 |
| | International government sources | 27 | 16.56 |
| Community | Irish community organisation resources (MPOWER, Man2Man) | 108 | 66.26 |
| | Community outreach workers | 13 | 7.98 |
| Healthcare professionals | Healthcare professional at a GP clinic | 8 | 4.91 |
| | Healthcare professional at a hospital | 9 | 5.52 |
| | Healthcare professional at a pharmacy | 0 | 0 |
| | Healthcare professional at a sexual health clinic | 34 | 20.86 |
| Social media | Twitter posts (non-governmental/health service) | 48 | 29.45 |
| | Facebook posts (non-governmental/health service) | 19 | 11.66 |
| | Instagram posts (non-governmental/health service) | 51 | 31.29 |
| | TikTok posts (non-governmental/health service) | 10 | 6.13 |
| | Hook-up/Dating apps | 74 | 45.4 |
| Other | Directly from friends | 38 | 23.31 |

**Table 3. Level of trust in information about mpox from different sources.**

| Source | | N | M | SD |
|---|---|---|---|---|
| Media | Irish TV/Radio | 150 | 3.45 | 1.11 |
| | International TV/Radio | 140 | 3.29 | 1.01 |
| Government | Irish government sources (HSE, Department of Health etc.) | 145 | 3.92 | 1.13 |
| | International government sources | 141 | 3.52 | 1.11 |
| Community | Irish community organisation resources (MPOWER, Man2Man) | 142 | 4.48 | 1.04 |
| | Community outreach workers | 132 | 3.76 | 1.00 |
| Healthcare professionals | Healthcare professional at a GP clinic | 133 | 3.82 | 1.18 |
| | Healthcare professional at a hospital | 133 | 3.94 | 1.08 |
| | Healthcare professional at a pharmacy | 132 | 3.30 | 1.19 |
| | Healthcare professional at a sexual health clinic | 133 | 4.47 | 1.00 |
| Social media | Twitter posts (non-governmental/health service) | 134 | 2.49 | 1.11 |
| | Facebook posts (non-governmental/health service) | 130 | 2.17 | 1.06 |
| | Instagram posts (non-governmental/health service) | 135 | 2.44 | 1.14 |
| | TikTok posts (non-governmental/health service) | 132 | 2.21 | 1.06 |
| | Hook-up/Dating apps | 138 | 3.17 | 1.09 |
| Other | Directly from friends | 131 | 2.99 | 0.90 |

participant. Sources varied widely: 25.15% used Irish TV/Radio, 10.43% used international TV/Radio, 16.56% used international government sources, and 23.31% obtained information directly from friends. Among healthcare professionals, those at sexual health clinics were most commonly consulted (20.86%), followed by professionals at hospitals (5.52%) and GP clinics (4.91%).

The trustworthiness of information sources varied significantly among participants (see Table 3). Social media sources were generally rated as less trustworthy, with Facebook posts being considered the least trustworthy (M = 2.17 out of 5). Trustworthiness appeared to increase with sources more directly related to the mpox public health emergency. Irish community-led organisations for gbMSM, were rated as extremely trustworthy (M = 4.48), followed closely by healthcare professionals in sexual health clinics (M = 4.47), and then Irish government sources, including the Health Service Executive (HSE) and Department of Health (M = 3.92).

Overall, participants felt moderately well-informed about the mpox public health emergency (M = 3.47) and believed they understood public health guidance reasonably well (M = 3.48). Participants' levels of concern about mpox (M = 2.78) and perceived risk of contracting mpox (M = 2.4) were lower than their concern about other STIs (M = 3.2) and perceived risk of contracting other STIs (M = 2.85), as shown in Table 4. This perception likely

**Table 4. Perceptions of aspects of mpox.**

| | N | M | SD |
|---|---|---|---|
| Concern about contracting mpox | 163 | 2.78 | 1.**25** |
| Concern about contracting other STIs | 163 | 3.20 | 1.15 |
| Perceived risk of contracting mpox | 163 | 2.40 | 1.02 |
| Perceived risk of contracting other STIs | 163 | 2.85 | 1.10 |
| Perceived negative impact of mpox infection | 157 | 3.86 | 1.27 |
| Actual negative impact of mpox infection | 5 | 3.20 | 2.05 |
| Perceived level of informedness about mpox public health emergency | 161 | 3.47 | 1.37 |
| Perceived level of understanding of public health guidance regarding mpox | 161 | 3.48 | 1.38 |

reflects the actual risk of infection for mpox compared to other STIs indicating that the participant's perceived level of knowledge was accurate. Despite this, participants reported a somewhat high perceived negative impact of mpox (M = 3.86), while those diagnosed with mpox reported a slightly lower negative impact (M = 3.2), albeit from an extremely small sample.

Based on the qualitative data elicited from the open questions and free text responses, we developed four distinct themes described below. The first theme highlights the perceived divergence across how the public health system, community organisations and community members responded to mpox. The second and third deal with issues of stigma, in how both mpox itself, but also how the PHEIC was responded to in Ireland were sources of stigma. The final theme the deals with how experiences of mpox elicited memories of other culturally significant health scares.

## Perceptions of the mpox response: Divergence in urgency, priority and care

Participants put forward a variety of views on how mpox was managed; broadly categorised into the formal statutory response, including government, the health service and public health system; the community organisation response; and the less formal or organised response across social networks.

Overall, there was disappointment and frustration with how the formal and statutory response was managed. Participants considered there to be a lack of urgency in the state's response, confounded by already poor sexual health infrastructure and an information vacuum. The following reflection is from one of the participations who expereinced a diagnosis of mpox

> We were very early cases in Ireland. . . Public health did not get in touch with us when we were tested. We called in covid to work. The week was over and symptoms raging and a week later still no sign of public health interaction. Then case was confirmed 8 days after testing. Public health took us out of work and ordered bedroom isolation for us. Complicated by having housemates. Housemates were vaccinated. Then isolation continued until cleared by GP. The over arching feeling was pure fear as the information vacuum at the start was total. We were tested and the result was hampered by the fact the sti clinic only worked certain days. There was no urgency because the clinic was closed. The procedures were new of course however with covid having happened I would have thought that guidelines for emerging infections would have been tighter. Much tighter. (Gay man, Westmeath, 36–40)

For many of the participants, this perceived lack of urgency in the management of mpox was related to an overall lack of care for the community by the state and its agencies, a form of institutionalised homophobia.

> The government response has been incredibly disappointing and has eroded any trust I had that they care about the LGBTQ+ community. The HSE response has met my incredibly low expectations—it is a shameful organisation and this is just a one more failure to add to a long list of them. (Gay Man, 31–35, Dublin)

> It really does feel like they [HSE] don't care about it especially compared to COVID, and I can't help feeling that part of that is because it's mainly affecting gay and bisexual men. Yes it's not as prevalent or deadly as COVID, but it's still a big issue, and it feels like nobody is acknowledging that. (Bisexual Man, 21–25, Dublin)

Much of the frustration reported in this survey was linked to the lack of availability and access to vaccination–there was an acknowledgement by some participants that this was an overall system failing and linked to global vaccine shortages, but comparisons were made to formal responses and vaccination programmes in other jurisdictions which appeared better.

Well as I work in the Healthcare community it's not taken seriously, don't mean that in a bad way, we were diagnosed in the UK Liverpool, there appears to be no procedure in Ireland to deal with it (Gay man, 56–60, Dublin)

Very poorly managed as New York State has been offering vaccine boosters for monkeypox before the initial dose was even available in Ireland (Gay Man, 26–30, Dublin)

When reflecting on the response, participants were clear to make a distinction between what was seen as the statutory response by government and its agencies and the response led by community organisations, which was seen to be very effective

Where is the public information campaign. Local groups, especially in the LGBT+ community are doing trojan work, but this should be a government led response to what's supposed to be a public health emergency (Gay Man, 21–25 Dublin)

No support from central government in the form of supports to those needing to self isolated, lacklustre response from HSE both in terms of information and vaccination. Very grateful to organisations like Man2Man and MPower for acting to inform and protect the community. (Gay Man, 36–40, Kildare)

Overall, the participants were very complimentary of the role of community organisations and activists in keeping members of the community informed and protected–this was expressed as being in contrast rather than complementary to the government and statutory response.

. . .the MPower reach in general is a fantastic initiative and without them I would have been unlikely to visit a sexual health doctor in the past. Their campaigning on Grindr about mpox should be award winning, it is colourful, interesting, to the point but isn't intrusive (as in you MUST click this ad which invariably means one won't click on it). I specifically remember an ad last week whereby it mentioned vaccine dates and locations at short notice! That was a brilliant initiative (Gay Man, 31–35, Laois)

There was also evidence in the responses to the survey, that as well as more formalised support, individual members of the community were responsive to caring for and informing each other in an informal way. The concerns about mpox were not just centred around its impact on individuals but also their friends, lovers and the wider community

I reached out to my friends and lovers once I saw the vaccine was being released on a self referral basis, I wanted to make sure everyone was looking after themselves sexually (Gay man, 31–35, Galway)

The LGBTQ community were open to being informed and to informing others (Gay man, 21–25, Dublin)

The concerns about what mpox would mean for the wider community were articulated by many participants and intersected with issues of stigma, otherness and discrimination.

## The mpox outbreak as a sign of otherness for gbMSM

The outbreak of mpox in Ireland was often discussed in ways that positioned it as a new source of stigma for gbMSM. This elicited fear and concern amongst participants, not necessarily around the impact of the disease on themselves as individuals, but rather on how this stigma would affect both wellbeing in their community and concern regarding mpox in the wider community. For this reason, many resisted the framing of mpox as a gay disease.

> In the early days, the messaging of 'no need for the general public to be worried because MPX primarily affects gay men and MSM' was unhelpfully stigmatizing. (Gay man, 36–40, Dublin)

> [I] Feel it has been portrayed in Ireland as another Gay plague. (Gay man, 56–60, Wexford)

The othering anticipated and experienced by participants was complex and layered across various intersecting aspects sources of stigma: the labelling of mpox as a sexually transmitted infection, the initial decision to prioritise those who had previously been diagnosed with early infectious syphilis, the lack of certainty about the characteristics of the disease, the visible nature of how the disease can affect one's appearance, and the association of mpox with moralistic, sex-negative narratives.

> [I'm] becoming a little more paranoid of judgements based around monkeypox [that] others may make of me if I openly disclose my orientation to new people. (Gay man, 18–20, Galway)

> Akin to coughing or sore throat symptoms in public during Covid, I became aware of my skin and appearance during this time. I have friends who have various skin conditions who were treated badly or asked prying questions during the summer [at the] height of symptoms and [when there was] no access to vaccines, in particular. (Gay man, 31–35, Dublin)

> [I'm] annoyed and angry with how the vaccine programme looks for specific people under specific conditions which when read by [the] wider public would suggest that those getting the vaccine could be of lower moral [standing]. (Gay man, 31–35, Dublin)

Many participants expressed worry that mpox would lead to situations in which their privacy would be invaded, and they would be exposed to a lack of discretion, confidentiality or understanding. These anxieties were linked to embarrassment in response to heteronormative understandings of sex between men and fuelled concerns around managing the disclosure of infection, given that sores from mpox may be difficult to conceal and the long isolation period would require extended absences from work and social events.

## The potential for othering through mpox prevention practices

The mpox outbreak also contributed to othering within communities of gbMSM; this was in part owing to the dominance of risk discourses in communication about the outbreak and vaccine eligibility. This focus on risk led to division among community members. Some responses framed gbMSM as lacking control, or as responsible for the mpox outbreak.

> I cannot help but theorise that the gay community lacks self-control or struggles to safely engage with others sexually (Gay man, 31–35, Dublin)

Several participants blamed others in the community for their perceived failure to support public health measures.

Given how promiscuous and careless many gay men are, plus the rampant use of drugs in sexual encounters, I have no confidence [. . .] that [mpox] will be contained (Bisexual man, 46–50, Dublin)

I think the gay saunas in Dublin should be shut down on public health grounds given that every friend, and friends of friends, seems to regularly contact STIs in them. (Bisexual man, 46–50, Kerry)

Judging the sex lives of others also led to informal risk assessments of potential partners to reduce the likelihood of acquiring mpox.

I have become very selective as to who I have sex with. (Gay man, 51–55, Donegal)

Blame and jealously were especially prominent in relation to vaccines; with participants suggesting that community members who acted to lower their risk were offsetting the behaviour of a high-risk subgroup.

the vaccines had to be prioritised to highest risk, but this indirectly punished those who actively chose to lower their risk exposure until the vaccine was available. (Gay man, 36, Dublin)

Competition for vaccination contributed to conflict among subgroups of community members, who argued that manipulation of the system by those of all risk levels limited their own access to vaccination. Some felt that those at lower risk had taken advantage of the vaccine system by misrepresenting their risk level.

Self-assessing risk on a website means a lot of people who are much lower risk have received the vaccine prior to genuinely high-risk individuals. (Gay man, 31–35, Dublin)

Other responses moralised the eligibility criteria. This, alongside competition among community members and risk categories, led to uncertainty around engaging with the vaccine programme.

You must self-identify as a slut to get [the vaccine] (Gay man, 46–50, Galway)

[The] HSE is offering it to high-risk people. [I] guess I'm at slight risk and maybe eligible but don't want to deprive others at higher risk than me. (Gay man, 46–50, Dublin)

This theme contrasts with the previously outlined positive informal response to mpox by community members, highlighting the complex social aspects of mpox. Division often resulted from issues of vaccine availability and fear of stigma from the public; our analysis of the ways in which participants articulated this fear is further developed in the final theme.

## mpox, memory and fear

The individual and collective emotional impact of mpox was significant for participants. While individual perceptions of the extent to which mpox was an issue varied, most participants expressed feelings of fear and concern. Key topics included the threat of potential infection, transmission to others, isolation, and the broader impact on the community.

Participants perceived infection as very serious, with descriptions of it as especially sore or painful evoking clear concern about contracting mpox and alterations to normal behaviour patterns. The impact extended beyond acute changes in behaviour, affecting social experiences and mental health.

> [A friend] described it as an excruciatingly painful experience. This really made me fearful, and completely prevented me from putting myself at risk for several months. (Gay man, 31–35, Dublin)

> My anxiety has skyrocketed again as I feel like I'm missing out on living a part of my life that brings me joy. (Gay man, 31–35, Dublin)

The fear of infection was exacerbated by uncertainty around prevention; while the quantitative data showed that participants generally felt they were well-informed, there were significant unknowns about this outbreak. Participants also expressed fear of spreading mpox to others, feeling a responsibility to protect loved ones and often reducing social and sexual contacts.

> I wouldn't know what the symptoms are or who to call or what to do if I get infected. (Gay man, 18–21, Dublin)

> I was very fearful of acquiring and passing it on to nephews, nieces, my elderly parents, and other (straight and gay) friends. (Gay man, 36–40, Dublin)

Concern for the broader community was also common with a cognisance of how loneliness is an issue within communities of gbMSM.

> I made sure to check in on friends who had contracted [mpox] or felt lonely because of the outbreak and fear leading to isolation. (Gay man, 36–40, Dublin)

Some participants were extremely vigilant about infection while others viewed mpox as an additional challenge amid existing issues.

> I'm very [paranoid] and looking at people's bodies and can't relax during sex. (Gay man, 36–40, Dublin)

> It's just another worry added to a larger bag of worries I have. (Gay man, 26–30, Dublin)

Concerns about isolation and its impact on work and mental health were significant. And participants felt unsupported and uncertain about the long-term health implications.

> One worry was the need to isolate for such a long time. This would have had a huge impact on my work with project deadlines to be met. I simply couldn't be away from work for 28 days. (Gay man, 46–50, Dublin)

> I'm still not sure if it will have any long-term implications for my health. For example, I've been exhausted since diagnosis over two months ago. It's left me feeling very upset and isolated. (Gay man, 61–65, Dublin)

Experiences of fear, and perceptions of mpox more broadly, were shaped by the memory–both individually and culturally–of previous culturally significant pandemics. There were many references the impact of HIV, and more recently COVID-19. Some reflected on the

impact HIV had on the community, and these fears were not limited to those who had lived through the HIV pandemic. Instead, they seemed to reflect the way memories of HIV have been embedded in LGBTQ+ culture.

> The Irish gay community is already so fragile, I really don't want this to be another AIDS crisis, I don't think my poor heart could take it. (Gay man, 21–25, Galway)

> It was difficult not to start thinking this would be the next AIDS/HIV. None of us want to have something like that hit our community ever again. (Gay man, 36–40, Dublin)

It seems these culturally significant pandemics made fear discourses regarding mpoxparticularly impactful, and the way respondents viewed mpox through a HIV lens likely exacerbated the many concerns they had. Some felt worried about the prospect of another illness being associated with gay men and were acutely aware of the potential social ramifications of this.

> Here we go again, gay men being blamed [for] another disease (Gay man, 51–55, Dublin)

> I worry about the stigma of monkey pox as much as anything else. I love to have sex, but this has driven me to abstinence. (Gay man, 46–50, Meath)

Others described the mpox emergency as being in some ways a continuation of COVID-19 and contrasted the reopening of wider society with the restrictions that persisted among gbMSM. Given that gbMSM can face structural barriers to health related to their identity, there may have been particular frustrations that others were allowed to return to life prior to the COVID-19 pandemic while gbMSM were once again left behind.

> It was hard at times to have to restrict hook-ups etc., because there was a fear mpox was inevitable, but at the same time I still wanted to enjoy the first normal summer in ages. (Gay man, 21–25, Dublin)

> [The mpox emergency] broadly coincided with socialising after Covid, felt like it was just a further obstacle to meeting people. (Gay man, 36–40, Dublin)

## Discussion

While the survey results suggest that broadly gbMSM received adequate information and support around mpox, negative views persisted about the public health response overall. It is important to note that the public health strategy developed by the crisis management and strategic advisory groups in Ireland was to facilitate the community organisation sector to take the lead and resource same. This divergence, with a very positive view of community organisations but disappointment and anger with the public health institutions may be linked to an overall perception of institutionalised homophobia within statutory health systems.

Individual negative experiences of mistreatment, discrimination or stigmatisation within the Irish health system may lead, not only to a dissatisfaction with healthcare received, but may also influence healthcare avoidance and a mistrust in health professionals and healthcare services [13]. While it is difficult to quantify exactly how these experiences might negatively impact on health outcomes, actual or anticipated perceptions and stigma have been shown to impact decisions around HIV and sexual health testing, which can have a subsequent effect on transmission [14].

While the community partnership approach demonstrated in Ireland, along with many other areas throughout the PHEIC is a model which undoubtedly has impact and should be encouraged, the negative perceptions of public health organisations expressed in this study is a cause for concern. Experience of homophobia within the health system is a common trend internationally [15] and needs attention if sexual minority communities are to build trust within the wider system. LGBTQ+ community organisations might play a key role in encouraging this trust building. In the context of disease outbreaks, there is a need for community partnership approaches to develop prevention and treatment strategies that reflect the social and cultural aspects of transmission, as too strong a focus on epidemiological dynamics can increase stigma, which itself can be a barrier to disease prevention in many ways.

While not all responses were heavily critical of the public health system and its explicit and visible response to the PHEIC, the fact that there were differing views suggests that a multiplicity of approaches to public health should be adopted.

Recognising and responding to the multifarious stigmatising features of the mpox outbreak was an early concern for affected communities, the community organisations that support them, public health officials, and healthcare practitioners globally. Despite this, public health measures that targeted specific sexual health histories among gbMSM were adopted in many jurisdictions, including Ireland, and this both individualised and stigmatised mpox further [16] as noted by our participants. This approach may have been motivated by a narrow focus on the epidemiological aspects of mpox to the neglect of its broader social implications [17].

The stigmatised nature of mpox meant that the impact of the outbreak reached further than those diagnosed with mpox and their loved ones and may have even reduced engagement with preventative measures [7]. Data from the Netherlands suggests that mpox stigma was mainly driven by concerns about it being seen as a gay disease or something for which gay men could be blamed [18]. As seen in our data, this was largely due to memories or awareness of HIV being framed in this way, and the negative attitudes towards sex between men at the root of HIV stigma [19].

Other dimensions to mpox stigma included its concentration of symptoms affecting the anogenital area, the visibility of skin lesions that may occur due to mpox, the racialised nature of blame for mpox due to its emergence in West Africa, and the potential disruption that a diagnosis of mpox can cause due to the long self-isolation period required to prevent onward transmission [20]. Logie [3] has linked these features of mpox to three stigma archetypes, namely the foreign other, the immoral other, and the visibly unwell other. Perhaps of most concern are reports of stigma and discrimination based on actual or perceived mpox diagnosis in healthcare settings [21], though this was not reported by our participants. Given its complex and pervasive nature, mpox stigma must be a key consideration in the development of any community health intervention to prevent mpox transmission or support those affected.

## Community collaboration for equitable public health interventions

The 2022 mpox outbreak had an inordinate impact on gay, bisexual and other men who have sex with men [22]. As a consequence, it was imperative that health services collaborated with these communities in order to deliver messaging to the general public that was culturally appropriate and non-stigmatising. Among participants in this study there were some conflicting views on whether this had been effective, and it has been highlighted as a significant consideration in evaluating the response of statutory agencies to the mpox outbreak in Ireland. This study affirms the importance of co-designed messaging, in partnership with at-risk communities, to promote effective context specific collaboration. This has also been demonstrated in other contexts where partnership with community organisations facilitated not only

enhanced communication strategies but also innovative delivery of healthcare interventions such as vaccination, testing and sexual health counselling within community spaces [7, 17, 20, 21, 23].

The concept of community collaboration has been an essential component of previous international public health outbreak response strategies. This was evident in the response to both Ebola and COVID-19 where communities fulfilled an important role in the development and application of appropriate and context specific prevention and control efforts [24]. However, where agencies overlook the values and preferences of communities, these efforts are undermined resulting in stigma, discrimination and a lack of trust [25].

In the context of the 2022 global mpox outbreak, it was essential to recognise the impact of structural inequalities to deploy equitable public health interventions for marginalised communities whilst emphasising the significance of community engagement and culturally sensitive interventions [26]. The potentially negative impact of underlying structural inequities, regarding public health response efforts, upon marginalised communities are stark and evident in how this PHEIC was experienced and responded to [23, 26, 27]. Once again, this is further compounded by adverse factors that include homophobia, stigma, and discrimination that were propagated on certain social media platforms during the mpox outbreak [23, 27].

In general, the findings of this study demonstrate an overall positive experience of gbMSM during the mpox outbreak in terms of information and support, specifically reflected in the collaborative approaches that were engendered between local and national public health agencies and community organisations. These collaborative efforts have been identified as effective based upon the communication, information and support received by individuals, and 'bottom-up' interventions to engage positively with community actors and at-risk populations. It is important that broader lessons are learned from these experiences to establish more robust and dynamic responses to future public health emergencies. If implemented appropriately, this type of engagement supports enhanced positive, emotional, psychological and physical connectedness between individuals and communities [28].

## Expanding sexual health definitions, services, prevention strategies

The research highlights the need to broaden the scope of sexual health definitions to encompass a more holistic understanding of gbMSM's experiences. This includes recognising the emotional and psychological impacts of health crises, as well as the intersection of gbMSM identities, sexual health and issues of stigma, otherness, and societal discrimination [29].

Moving from a position of sexual health to one of sexual wellbeing via the integration of mental health supports, pleasure-focused practice and addressing sexual stigmatisation explicitly could help create more supportive environments that acknowledge and address the multifaceted nature of gbMSM's health needs [30].

The stark contrast between the state's response and community-based initiatives, as reported by participants, underscores the urgent need for more robust and responsive sexual health services. The recent introduction of the free-to-access, nationally available at-home STI testing service has improved the availability of and accessibility to sexual health services in Ireland [31]. However, further development of these services to meet the needs identified in this research would involve improving the infrastructure and accessibility of sexual health clinics, ensuring they are resourced to operate beyond limited hours to provide timely sexual healthcare.

Prevention strategies must also evolve to be more inclusive and proactive. Participants' experiences of fear and uncertainty due to the information vacuum at the onset of the mpox outbreak point to the need for comprehensive, accurate, and timely public health information

that is effectively delivered. Social media is a valuable tool in the dissemination of public health information [32]. However, this research has highlighted caution must be used as trust in social media platforms can vary. It is also important to note that certain subgroups (e.g., older people, people with disabilities or migrants) are at risk of further marginalisation if there is an overreliance on social media communications [32]. Prevention efforts should prioritise clear communication about routes of transmission, symptoms, and preventive measures, ensuring that these messages do not inadvertently stigmatise specific groups [33]. Additionally, the criteria for vaccination and other preventive measures should be transparent and equitable, avoiding perceptions of bias or blame based on sexual behaviour [34].

Finally, addressing the internal othering within the gbMSM community, as noted in the study, requires prevention strategies that promote solidarity rather than division. Public health campaigns should emphasise collective responsibility and community support, highlighting that when managing a disease outbreak, everyone's health is interconnected.

## Limitations

Although using an online cross-sectional survey was necessary to ensure a broad range of participants within a short period, it introduced some limitations. The cross-sectional nature of the survey meant that respondents' answers could not be further probed beyond the information they provided, potentially leading to misinterpretation of some data. Despite the advisory group's assessment of the survey questions' phrasing, the scales used to collect quantitative data were not validated.

The results of this study represent the views of a sample of gay, bisexual, and other men who have sex with men (gbMSM) in Ireland between December 2022 and January 2023. While this provided a useful overview of this group's experiences, perceptions of mpox likely varied throughout the outbreak in response to incidence numbers and the public health response. Additionally, the sample was recruited through social media and posters in LGBT+ venues, likely overrepresenting participants engaged with LGBT+ community social networks. The survey materials were primarily distributed by the MPOWER programme at HIV Ireland, which may have skewed the sample toward those engaged with the service. Consequently, participants living in Dublin (57.06%), ethnically Irish participants (84.66%), those identifying as gay (85.28%), degree holders (73.62%), and those with higher incomes (70.55% earning more than €40,000) are overrepresented. Therefore, our data do not necessarily provide insight into how mpox interacted with experiences of financial hardship or identity-related marginalization. Data analysis also included only five (3.07%) participants who had contracted mpox, limiting the claims based on these experiences.

## Conclusion

The 2022 mpox outbreak highlighted critical areas in public health response, particularly concerning marginalised communities such as gbMSM. Our study demonstrates that community-based organisations played a pivotal role in providing information and support, reflecting the effectiveness of collaborative efforts between public health agencies and these communities. However, the outbreak also underscored significant challenges, including stigmatisation, discrimination, and mistrust towards public health institutions, potentially influenced by experiences of institutionalised homophobia.

While community-led initiatives were positively received in the Irish context, there remains a crucial need to address the broader structural inequalities and stigma that exacerbates public health crises. Public health strategies must evolve to be more inclusive, culturally sensitive, and transparent, ensuring equitable access to information, vaccination, and care. Additionally,

redefining sexual health to encompass mental well-being and addressing stigma explicitly can create more supportive environments for gbMSM.

The lessons from this PHEIC emphasise the importance of robust community collaboration in public health interventions. Such approaches not only enhance the efficacy of responses to health emergencies but also foster greater trust and engagement within affected communities. Actionable recommendations may include the establishment of regular community forums between public health bodies and marginalised communities and further development of community-based healthcare initiatives. The examples of collaboration demonstrated through the PHEIC, such as outreach events, community-based vaccination and testing, could potentially be mainstreamed into more general delivery of public health initiatives.

While throughout 2023 and 2024 cases of mpox significantly decreased internationally amongst communities of gay and bisexual men, the disease remains endemic in several regions across central and western Africa, with ongoing transmission outside of endemic countries [35]. A new PHEIC was declared in 2024 due to a more severe clade of the virus circulating in the Democratic Republic of Congo with a small number of cases being reported in other regions [36].The lack of attention given to mpox historically in these regions was the very issue precipitating the PHEIC in 2022. Infectious diseases do not respect borders, and if we are to have a truly equitable and effective public health prevention strategy, it must be on a truly global scale [37]

As we move forward, it is imperative to integrate these insights into future public health strategies, ensuring that responses are dynamic, equitable, and attuned to the diverse needs of all populations globally.

## Supporting information

**S1 Appendix. Survey questions.**
(DOCX)

## Author Contributions

**Conceptualization:** John Gilmore, David Comer, David J. Field, Adam Shanley, Chris Noone.

**Data curation:** John Gilmore, David Comer, Chris Noone.

**Formal analysis:** John Gilmore, David Comer, Chris Noone.

**Funding acquisition:** John Gilmore, Adam Shanley, Chris Noone.

**Investigation:** John Gilmore, David Comer, Chris Noone.

**Methodology:** John Gilmore, David Comer, Chris Noone.

**Project administration:** John Gilmore, David Comer, Chris Noone.

**Supervision:** John Gilmore, Chris Noone.

**Validation:** John Gilmore, David J. Field, Randal Parlour, Adam Shanley, Chris Noone.

**Visualization:** John Gilmore, Chris Noone.

**Writing – original draft:** John Gilmore, David Comer, David J. Field, Randal Parlour, Chris Noone.

**Writing – review & editing:** John Gilmore, David Comer, David J. Field, Randal Parlour, Adam Shanley, Chris Noone.

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
