## [Decision Letter · Decision Letter 0]

16 Sep 2024

PONE-D-24-32402Recognising and responding to community needs of gay and bisexual men around mpoxPLOS ONE

Dear Dr. gilmore,

Thank you for submitting your manuscript to PLOS ONE. After careful consideration, we feel that it has merit but does not fully meet PLOS ONE’s publication criteria as it currently stands. Therefore, we invite you to submit a revised version of the manuscript that addresses the points raised during the review process.

Editor suggestion:Please improve the language of the manuscript.  Comments from PLOS Editorial Office: We note that one or more reviewers has recommended that you cite specific previously published works. As always, we recommend that you please review and evaluate the requested works to determine whether they are relevant and should be cited. It is not a requirement to cite these works. We appreciate your attention to this request.

We look forward to receiving your revised manuscript.

Kind regards,

Ranjan K. Mohapatra, PhD

Academic Editor

PLOS ONE

Journal Requirements:

“Funding from Health Services Executive Sexual Health and Crisis Pregnancy Programme”

5. We noted in your submission details that a portion of your manuscript may have been presented or published elsewhere. [Inital research results are published on the HIV Ireland website in the form of a short report. Discussion has been developed and findings contextualised in the submission to PLOS ONE] Please clarify whether this [conference proceeding or publication] was peer-reviewed and formally published. If this work was previously peer-reviewed and published, in the cover letter please provide the reason that this work does not constitute dual publication and should be included in the current manuscript.

6. In this instance it seems there may be acceptable restrictions in place that prevent the public sharing of your minimal data. However, in line with our goal of ensuring long-term data availability to all interested researchers, PLOS’ Data Policy states that authors cannot be the sole named individuals responsible for ensuring data access (http://journals.plos.org/plosone/s/data-availability#loc-acceptable-data-sharing-methods).

Reviewers' comments:

Reviewer's Responses to Questions

**Comments to the Author**

1. Is the manuscript technically sound, and do the data support the conclusions?

Reviewer #1: Yes

Reviewer #2: Yes

Reviewer #3: Yes

2. Has the statistical analysis been performed appropriately and rigorously? 

Reviewer #1: I Don't Know

Reviewer #2: Yes

Reviewer #3: N/A

3. Have the authors made all data underlying the findings in their manuscript fully available?

Reviewer #1: Yes

Reviewer #2: Yes

Reviewer #3: No

4. Is the manuscript presented in an intelligible fashion and written in standard English?

Reviewer #1: Yes

Reviewer #2: Yes

Reviewer #3: Yes

5. Review Comments to the Author

Reviewer #1: Although your article (PONE-D-24-32402) is organised effectively, there are a few places where it could be more focused, succinct, and clear. Based on the overall composition, the following particular suggestions will improve your document: I recommend this manuscript after minor revision.

1. Consistency in Title and Terminology:

• Ensure that the language is consistent. As an illustration, you utilise both "mpox" and "mbpox." Stick to a single term throughout the entire document. Per the most recent WHO standards, the recommended term is "mpox".

2. Context: Explain why the mpox outbreak in 2022 worried gbMSM populations especially. This should include a succinct description of the virus's characteristics, the reasons certain communities were disproportionately afflicted, and any early opinions or responses from the general public.

Relevance: Highlight why the study is specifically focused on Ireland. Is it due to a unique public health response, or were there specific community dynamics in play?

3. Methods

• Detail the Methods: You could provide a little more detail to the description of your mixed-methods approach, like the duration of the survey, the participant recruitment process, and any analysis tools that were employed.

• Reflexive Thematic Analysis: Mention briefly what "reflexive thematic analysis informed by critical realism" entails, especially for readers who may not be familiar with these concepts.

4. Findings

• Contextualize the Numbers: While stating that 163 men participated, it could be useful to provide some context, like how representative this sample is of the wider gbMSM community in Ireland.

• Thematic Overview: You’ve listed the four themes developed from the qualitative data. Consider briefly explaining what each theme entails or its significance rather than just listing them. This gives the reader a better grasp of the findings.

5. Discussion

• Strengthen the Argument: Although you discuss the efficacy of community-led projects, it would be helpful to give specific instances or further describe these activities. This would support your claim about the significance of these kinds of projects.

• Institutional Critique: Any particular instances from the research or literature that highlight the negative effects of institutionalised homophobia on public health outcomes could be added to the conversation about the issue.

• Balance the Discussion: In addition to stressing the difficulties, it could be a good idea to highlight any achievements or compliments received on the public health response, if any. This makes the conversation more fair-minded.

6. Conclusion

• Actionable Recommendations: Your conclusion is strong, but it would benefit from more actionable recommendations. For instance, how can public health strategies are made more inclusive and transparent? What specific steps can be taken to ensure equitable access and address structural inequalities?

7. Language & Tone

• Conciseness: There are some opportunities to tighten the language. For example, "the study underscores the need for inclusive, culturally sensitive, and transparent public health strategies" could be simplified to "the study highlights the need for inclusive and transparent public health strategies."

• Avoid Jargon: Some readers may not be familiar with terms like "critical realism" or "reflexive thematic analysis." While these are important, consider simplifying or explaining them briefly to ensure accessibility.

8. Overall Structure

• Flow & Transitions: Ensure smooth transitions between sections. For example, the findings section should naturally lead into the discussion, where the implications of these findings are explored in more depth.

Incorporating these improvements should make your write-up clearer, more persuasive, and accessible to a broader audience.

Reviewer #2: This paper reports on a community-based survey of gay and bisexual men in Ireland on the 2022 mpox outbreak. The paper employs a mixed methods analysis, which suggests that gay and bisexual men were well informed about mpox, but were critical of the government’s response and highly concerned about stigma. The authors suggest that robust community collaboration and ensuring equitable responses are vital.

This is an excellent paper; the authors have done a fantastic job of presenting in-depth findings from a relatively small community sample, including effectively presenting qualitative findings in a layered and sophisticated way from survey responses that in my experience, can often be done in a way that is quite ‘shallow’. I’ve also seen articles on mpox from much larger sample sizes that really lack the depth and resonance displayed here. There is an excellent alignment between the aims, methods, analysis, and discussion. Fantastic job!

My suggested revisions below are very minor, e.g., typographical, adding a little bit more detail in places, and some small suggestions of other sources for the discussion.

MINOR SUGGESTED REVISIONS:

Abstract

Minor spelling error – ‘mbpox’ should be ‘mpox’?

Introduction

Minor spelling error – as an institution, the spelling is ‘World Health Organization’

Should the authors mention somewhere that the WHO declared this outbreak no longer a PHEIC in 2023? (noting that there’s currently a different outbreak declared a PHEIC).

Methods

Participants – could the authors distinguish which recruitment methods applied to the convenience sampling and purposive sampling, respectively?

I think you could provide a brief descriptive summary of the participants here to complement Table 1. E.g., ‘the majority of participants identified as gay, the average age group, etc.’

Findings

In the first theme (divergence), it appears that the first quote (Gay man, Westmeath, 36-40) is from a person who was diagnosed with mpox from the context of the quote? I think this could be made explicit before you introduce the quote.

The second theme is interesting – from what I saw there were claims about stigma in both directions. E.g., framing mpox as a ‘gay disease’ being stigmatising, but also that saying ‘everyone is at risk’ was inappropriate for risk messaging and resource allocation, which constituted a different form of stigma from the state. It makes me think of this op ed:

Highleyman, L. (2022, July 25). Not Everyone Has to Be Equally Worried About Monkeypox. Slate. https://slate.com/technology/2022/07/monkeypox-spread-sex-men-vaccines-worry.html

Some similar themes in this paper expressed in this chapter and article. E.g., comparisons to AIDS crisis and COVID-19, the framing of risk groups, frustrations with public health response, concerns for embedded community connections:

https://www.taylorfrancis.com/chapters/oa-edit/10.4324/9781003322788-10/thinking-hiv-pandemic-times-kiran-pienaar-dean-murphy

Storer, D., Holt, M., Paparini, S., Haire, B., Cornelisse, V. J., MacGibbon, J., Broady, T. R., Lockwood, T., Delpech, V., McNulty, A., & Smith, A. K. J. (2024). Informed, but uncertain: Managing transmission risk and isolation in the 2022 mpox outbreak among gay and bisexual men in Australia. Culture, Health & Sexuality, 1–16. https://doi.org/10.1080/13691058.2024.2346540

General comment:

I noticed a switch to using MPOX (all capitalised) at one stage, which was inconsistent with earlier points in the article where it was talked about as mpox.

Reviewer #3: Many thanks for the opportunity to review this manuscript. This is an important, topical and timely piece of work that I enjoyed reading It would be helpful if the survey could be included as an appendix. The majority of content in the paper appears to be reporting the free text answers – so it would be good to see what the questions were. The results are really interesting to read – so much seems to be influenced by the post COVID (and HIV) context. During the 2022 Mpox outbreak I was able to observe how communities harnessed the power of social media to connect and support each other (e.g., informing others about vaccine availability etc). It’s great to see that this was something that was actively supported in Ireland. I completely agree with the conclusion that better collaboration between community organizations and government are needed (as well as better, and more funding for sexual

6. PLOS authors have the option to publish the peer review history of their article (what does this mean?). If published, this will include your full peer review and any attached files.

Reviewer #1: **Yes: **Dr. Puneet Kumar Singh

Reviewer #2: No

Reviewer #3: No

---

## [Author Response · Author response to Decision Letter 0]

20 Sep 2024

Reviewer 1

1. We have addressed the stylistic inconsistencies and errors throughout the manuscript as advised

2. We have given some further detail on why this outbreak was particularly of concern to communities of gbMSM including reference to its transmission characteristics. We have also given further information around the Irish context to contextualise this study.

3. Information regarding the duration of the study and the tools used are contained in the manuscript and these were made clearer. We have given further explanation of the terms and process of critical realism and reflexive thematic analysis. 

4. We have clarified that there is no census data which addresses sexual orientation in Ireland so it is difficult to comment on how representative the sample is, but that a diversity in demographics were represented. We have also given an overview of the themes as advised. 

5. We have noted the types of community-engaged initiatives represented in the papers cited to support our discussion on the important role they can play. We have added in some research around how stigma has been shown to impact on HIV and sexual health testing likelihood amongst gbmsm. We have also clarified that not all responses heavily criticised the public health response from the statutory services, highlighting the need for a variety or response modes.

6. We have noted some more actionable recommendations to bring the findings forward.

7. We have retained the term ‘culturally sensitive’ as we felt it was important to specify this.

8. We have addressed some of the language and tone throughout the manuscript to help the flow.

Reviewer 2

1. We have addressed the stylistic inconsistencies and errors throughout the manuscript as advised

2. We have noted in the conclusion that there has been a subsequent outbreak leading to another PHEIC being announced by the WHO and contextualised this in relation to the wider discussion. 

3. We have specified the sampling strategies

4. We have given a brief description of the sample (full details remain in the table)

5. We have made clear the specific quote referred to is taken from the response of a participant who had experienced mpox.

Reviewer 3

We have included the survey questions as an appendix.

---

## [Decision Letter · Decision Letter 1]

23 Oct 2024

Recognising and responding to community needs of gay and bisexual men around mpox

PONE-D-24-32402R1

Dear Dr. gilmore,

We’re pleased to inform you that your manuscript has been judged scientifically suitable for publication and will be formally accepted for publication once it meets all outstanding technical requirements.

Kind regards,

Ranjan K. Mohapatra, PhD

Academic Editor

PLOS ONE

Additional Editor Comments (optional):

Reviewers' comments:

Reviewer's Responses to Questions

**Comments to the Author**

1. If the authors have adequately addressed your comments raised in a previous round of review and you feel that this manuscript is now acceptable for publication, you may indicate that here to bypass the “Comments to the Author” section, enter your conflict of interest statement in the “Confidential to Editor” section, and submit your "Accept" recommendation.

Reviewer #1: All comments have been addressed

Reviewer #2: All comments have been addressed

Reviewer #3: All comments have been addressed

2. Is the manuscript technically sound, and do the data support the conclusions?

Reviewer #1: Yes

Reviewer #2: Yes

Reviewer #3: Yes

3. Has the statistical analysis been performed appropriately and rigorously? 

Reviewer #1: N/A

Reviewer #2: Yes

Reviewer #3: N/A

4. Have the authors made all data underlying the findings in their manuscript fully available?

Reviewer #1: No

Reviewer #2: Yes

Reviewer #3: Yes

5. Is the manuscript presented in an intelligible fashion and written in standard English?

Reviewer #1: Yes

Reviewer #2: Yes

Reviewer #3: Yes

6. Review Comments to the Author

Reviewer #1: (No Response)

Reviewer #2: The authors have addressed my feedback, which were minor. Great article!

I'm apparently supposed to write a minimum of 100 characters here, got to love these silly systems.

Reviewer #3: (No Response)

7. PLOS authors have the option to publish the peer review history of their article (what does this mean?). If published, this will include your full peer review and any attached files.

Reviewer #1: No

Reviewer #2: No

Reviewer #3: No

---

## [Editor Report · Acceptance letter]

25 Oct 2024

PONE-D-24-32402R1 

PLOS ONE

Dear Dr. Gilmore, 

I'm pleased to inform you that your manuscript has been deemed suitable for publication in PLOS ONE. Congratulations! Your manuscript is now being handed over to our production team.

Kind regards, 

on behalf of

Dr. Ranjan K. Mohapatra 

Academic Editor

PLOS ONE